# Biomimetic Collagen Membranes as Drug Carriers of Geranylgeraniol to Counteract the Effect of Zoledronate

**DOI:** 10.3390/biomimetics9010004

**Published:** 2023-12-22

**Authors:** Francisco Javier Manzano-Moreno, Elvira de Luna-Bertos, Manuel Toledano-Osorio, Paula Urbano-Arroyo, Concepción Ruiz, Manuel Toledano, Raquel Osorio

**Affiliations:** 1Biomedical Group (BIO277), Department of Stomatology, School of Dentistry, University of Granada, 18071 Granada, Spain; fjmanza@ugr.es; 2Instituto Investigación Biosanitaria, ibs. Granada, 18012 Granada, Spain; crr@ugr.es (C.R.); toledano@ugr.es (M.T.); rosorio@ugr.es (R.O.); 3Biomedical Group (BIO277), Department of Nursing, Faculty of Health Sciences, University of Granada, 18016 Granada, Spain; 4Faculty of Dentistry, University of Granada, Colegio Máximo de Cartuja s/n, 18071 Granada, Spain; mantoled@ucm.es (M.T.-O.); paulaurbanoarroyo7@gmail.com (P.U.-A.); 5Institute of Neuroscience, University of Granada, Centro de Investigación Biomédica (CIBM), Parque de Tecnológico de la Salud (PTS), 18071 Granada, Spain

**Keywords:** zoledronate, collagen membranes, osteoblast, geranylgeraniol

## Abstract

To counteract the effect of zoledronate and decrease the risk of osteonecrosis of the jaw (BRONJ) development in patients undergoing guided bone regeneration surgery, the use of geranylgeraniol (GGOH) has been proposed. Collagen membranes may act as biomimetical drug carriers. The objective of this study was to determine the capacity of collagen-based membranes doped with GGOH to revert the negative impact of zoledronate on the growth and differentiation of human osteoblasts. MG-63 cells were cultured on collagen membranes. Two groups were established: (1) undoped membranes and (2) membranes doped with geranylgeraniol. Osteoblasts were cultured with or without zoledronate (50 μM). Cell proliferation was evaluated at 48 h using the MTT colorimetric method. Differentiation was tested by staining mineralization nodules with alizarin red and by gene expression analysis of bone morphogenetic proteins 2 and 7, alkaline phosphatase (ALP), bone morphogenetic proteins 2 and 7 (BMP-2 and BMP-7), type I collagen (Col-I), osterix (OSX), osteocalcin (OSC), osteoprotegerin (OPG), receptor for RANK (RANKL), runt-related transcription factor 2 (Runx-2), TGF-β1 and TGF-β receptors (TGF-βR1, TGF-βR2, and TGF-βR3), and vascular endothelial growth factor (VEGF) with real-time PCR. One-way ANOVA or Kruskal–Wallis and post hoc Bonferroni tests were applied (*p* < 0.05). Scanning electron microscopy (SEM) observations were also performed. Treatment of osteoblasts with 50 μM zoledronate produced a significant decrease in cell proliferation, mineralization capacity, and gene expression of several differentiation markers if compared to the control (*p* < 0.001). When osteoblasts were treated with zoledronate and cultured on GGOH-doped membranes, these variables were, in general, similar to the control group (*p* > 0.05). GGOH applied on collagen membranes is able to reverse the negative impact of zoledronate on the proliferation, differentiation, and gene expression of different osteoblasts’ markers.

## 1. Introduction

In recent years, the use of polymeric membranes in different medical applications has been evolved [1,2]. Biomimetic collagen membranes and scaffolding fabrication technologies that may imitate the microenvironment of natural extracellular matrices (ECM) have been developed for guided bone regeneration (GBR) and guided tissue regeneration (GTR) [3]. Their ability to mimic natural ECM providing a supportive environment for cell growth and tissue regeneration, and the fact that they are self-healing materials makes them a valuable resource for medical and dental professionals seeking to improve patient outcomes in a variety of applications [4].

One of the applications of biomimetic polymeric membranes in the biomedical field is represented by the development of drug delivery systems such as antibiotics or proteins [5]. Drug release can be achieved through diffusion of active substances loaded on the polymeric membrane, resulting in a sustained drug release [6]. Collagen-based membranes, are the most widely used in GBR/GTR procedures, mainly due to their biomimetic properties, biocompatibility, good handling and biological ability to attract and activate gingival fibroblasts, periodontal ligament cells, and osteoblasts [7]. Different substances have been used to functionalize these membranes, such as corticosteroids or antibiotics with the aim of improving their biological properties [8,9].

Bisphosphonates (BPs) are synthetic compounds analogous to pyrophosphates where the oxygen atom responsible for binding the phosphate groups is replaced by a carbon atom. These drugs are used to treat certain bone pathologies such as malignant hypercalcaemia, prostate, and breast cancer bone metastases, Paget’s disease and osteoporosis [10,11,12]. These drugs are classified into nitrogenous and non-nitrogenous BPs. Nitrogenous BPs, such as zoledronate, pamidronate, alendronate or ibandronate, inhibit the enzyme farnesyl diphosphate synthase and disrupt the mevalonate pathway. Consequently, geranylgeranylpyrophosphate (GGPP) production in the mevalonate pathway is reduced [13], leading to impaired osteoclastic function [10,14].

These drugs are widely used, and their ability to inhibit osteoclast formation and activity has been extensively tested [15,16]. However, the exact effect that they produce on osteoblasts and bone is not fully understood. Previous studies stated that high doses of bisphosphonates reduce osteoblast-like cell proliferation through cell cycle arrest and induce cell apoptosis/necrosis [17]. However, at low doses, BPs increase cell proliferation while inhibiting differentiation of this cell lineage [18,19].

BPs have been associated with the development of some BRONJ [20,21]. Even when therapeutic doses of BPs are low, after prolonged treatment, a high concentration of BPs may be found in bone. It is due to the strong binding of these drugs to hydroxyapatite [18].

Patients treated with BPs may need to undergo oral cavity rehabilitation procedures involving the use of osseointegrated implants and guided bone regeneration (GBR). It requires the use of a membrane in order to compartmentalize the bone defect and prevent its colonization by non-bone forming cells, allowing the regeneration of hard tissues that have been damaged or lost [22]. The biological mechanisms responsible for wound regeneration and healing have been studied, which has allowed polymeric membranes to move from being a barrier to playing a more active role by combining biological activity with a barrier effect to promote tissue regeneration [23].

Geranylgeraniol (GGOH) is a diterpenoid that is hexadeca-2,6,10,14-tetraene substituted by methyl groups at positions 3, 7, 11, and 15 and a hydroxy group at position 1. It has been proposed as a potential therapeutic weapon for the treatment and prevention of BRONJ. GGOH is converted into GGPP, which serves as a substrate for the geranylgeranylation of proteins [24,25,26]. It is a poor water-soluble drug [27]. The enhanced loading capacity of poor water-soluble drugs into amine-containing materials has been previously reported. It is supposed to be via a reaction with the amine of collagen chains, forming amide bonds [28].

So far, there are no studies analysing the effect of doping collagen membranes with GGOH on bisphosphonate-treated osteoblasts. This fact can be considered very relevant because there are many patients treated with bisphosphonates who will need to undergo GBR/GTR procedures for oral rehabilitation.

The objective of this study was to determine the capacity of collagen-based membranes doped with GGOH to revert the negative impact of zoledronate on the growth and differentiation of human osteoblasts.

## 2. Methods

### 2.1. Collagen Membranes Functionalization

Collagen membranes obtained from porcine pericardium (Jason^®^; Botiss Biomaterials GmbH, Berlin, Germany) were adapted to 7 mm diameter discs and doped with GGOH. For this process, an ethanol (Sigma, St. Louis, MO, USA), solution of GGOH 10 μM was prepared. An amount of 15 μL of this solution was added to each membrane disc. Hence, two groups of membranes were obtained: (1) undoped collagen membranes (Col-M), and (2) GGOH-doped collagen membranes (GG-M).

### 2.2. Cell Culture

Human osteoblastic osteosarcoma MG-63 cells (ATCC, Manassas, VA, USA) were cultured in Dulbecco’s modified Eagle’s medium (DMEM; Invitrogen Gibco Cell Culture Products, Carlsbad, CA, USA). DMEM medium was supplemented with gentamicin 50 mg/mL (Braum Medical SA, Jaén, Spain), amphotericin B 2.5 mg/mL (Sigma, St. Louis, MO, USA), amphotericin B 2.5 mg/mL (Sigma, St. Louis, MO, USA), glutamine 1% (Sigma, St. Louis, MO, USA) and HEPES 2% (Sigma, St. Louis, MO, USA), penicillin 100 IU/mL (Lab Roger SA, Barcelona, Spain), 2% HEPES (Sigma, St. Louis, MO, USA), and 1% glutamine (Sigma, St. Louis, MO, USA), and subsequently supplemented with 10% fetal bovine serum (FBS; Gibco, Paisley, UK). A humidified atmosphere at 37 °C with 95% air and 5% CO_2_ was used to maintain the cultures. A solution with 0.05% trypsin (Sigma, St. Louis, MO, USA) and 0.02% ethylenediaminetetraacetic acid (EDTA; Sigma, St. Louis, MO, USA) was used to detach the cells from the culture flask. After this, the cells were resuspended in DMEM medium containing 10% FBS [29]. MG-63 cells were seeded onto both groups of collagen membranes (Col-M and GG-M) and cultured with or without the presence of a nitrogen-containing BP, zoledronate (Sigma-Aldrich, St. Louis, MO, USA) at a dose of 50 μM for 48 h.

### 2.3. Cell Proliferation Assay

Cells were seeded onto collagen membranes at a density of 1 × 10^4^ cells/mL, in a 24-well plate under the culture conditions described above. After 48 h, cell proliferation was analysed using the 3-(4,5-dimethylthiazol-2-yl)-2,5-diphenyltetrazolium (MTT) assay. In this assay DMEM medium is replaced by DMEM without phenol red with MTT 0.5 mg/mL (Sigma, St. Louis, MO, USA), incubated for 4 h, after which insoluble formazan crystals from the cell reduction of MTT are dissolved by the addition of dimethyl sulfoxide (Merck Biosciences, Darmstadt, Germany), and a spectrophotometer (Sunrise, Tecan, Männedorf, Switzerland) was used to read the absorbance at 570 nm [30]. The experiments were run in triplicate.

### 2.4. RNA Extraction and Real-Time Quantitative Polymerase Chain Reaction (RT-qPCR)

RNA from osteoblastic cells was extracted after 48 h of culture using the Qiagen RNeasy extraction kit (Qiagen Inc., Hilden, Germany), and a UV spectrophotometer at 260 nm (Eppendorf AG, Hamburg, Germany) was used to measure the amount of mRNA. An amount of 1 μg of mRNA was obtained for each of the examples and brought to 40 μL volume, after which it was reverse transcribed into cDNA to be amplified using the iScriptTM kit (Bio-Rad Laboratories, Hercules, CA, USA) by PCR technique [19]. Primer3-design software was then used to design primers to detect the mRNA of the following genes: bone morphogenetic proteins 2 and 7, alkaline phosphatase (ALP), bone morphogenetic proteins 2 and 7 (BMP-2 and BMP-7), type I collagen (Col-I), osterix (OSX), osteocalcin (OSC), osteoprotegerin (OPG), receptor for RANK (RANKL), runt-related transcription factor 2 (Runx-2), TGF-β1 and TGF-β receptors (TGF-βR1, TGF-βR2, and TGF-βR3), and vascular endothelial growth factor (VEGF). Peptidylprolyl isomerase A (PPIA), ubiquitin C (UBC) and ribosomal protein S13 (RPS13) were used as housekeeping genes to normalize the results [31,32]. The primer sequences are included in Table 1.

SsoFast^TM^ EvaGreen^®^ Supermix kit (Bio-Rad Laboratories, Hercules, CA, USA) was used for RT-qPCR. A total of 5 μL of cDNA was obtained for each sample and plated in 96-well plates for amplification using a thermal cycler (IQ5-Cycler, Bio-Rad Laboratories, Hercules, CA, USA). Temperatures were set at 60–65 °C and 72 °C for annealing and elongation, respectively, for more than 40 cycles. A standard curve for the target genes was obtained by plotting Ct values against log cDNA dilution. This was followed by agarose gel electrophoresis to create a melting profile. The proportion of ng of mRNA per average ng of housekeeping mRNA was calculated. The experiments were run in triplicate.

### 2.5. Nodules Formation and Matrix Mineralization

Alizarin Red S technique was applied to detect calcium deposits in the cell matrix [33]. Osteoblasts were seeded at a density of 5 × 10^4^ cells/mL per well on collagen membranes and cultured in osteogenic medium (DMEM without gentamicin and penicillin supplemented with 5 mM β-glycerophosphate and 0.05 mM ascorbic acid) with 400 μg/mL amoxicillin or 150 μg/mL clindamycin. Matrix mineralization analysis was carried out at 15 and 21 d of culture, capturing digital images of the staining. Cetylpyridinium chloride 10% (*w*/*v*) was used for 15 min to dissolve the red calcium deposits. Finally, the absorbance of the extracted stain was measured with a spectrophotometer at 562 nm (Biotek EL×800)

### 2.6. Scanning Electron Microscopy (SEM)

After osteoblasts were cultured on membrane disks in a 24-well plate at a density of 1 × 10^4^ cells/mL for 48 h, these membranes were subjected to critical point drying after fixation and then covered with carbon to evaluate cell morphology with a field emission scanning electron microscope (FESEM) (GEMINI, Carl Zeiss SMT, Oberkochen, Germany) [34].

### 2.7. Statistical Analysis

Normality was tested using the Kolmogorov–Smirnov test, after which comparisons between experimental groups were made using the one-way ANOVA test for variables following a normal distribution or by Kruskal–Wallis. The Bonferroni post hoc test was then applied for multiple comparisons. Data were expressed as mean and standard deviation, with significance set at *p* < 0.05.

## 3. Results

### 3.1. Cell Proliferation Assay

The attained mean and standard deviations of the osteoblastic cell proliferation in the MTT assay are presented in Figure 1. When the osteoblasts were treated with 50 μM zoledronate (which is reported as a therapeutic dose [18,19,24]) in the undoped (Col-M) group, a significant decrease in cell proliferation of about 65% (*p* < 0.001) was observed if compared to untreated cells. When osteoblasts were treated with zoledronate 50 μM and grown on the GG-M group, proliferation was two-fold higher if compared to the undoped membranes (*p* < 0.001). In the absence of zoledronate, no difference was found in proliferation between the control (Col-M) and the GG-M groups (*p* > 0.05).

### 3.2. Real-Time Quantitative Polymerase Chain Reaction

#### 3.2.1. Gene Expression of TGF-β1 and Its Receptors (TGF-β R1, TGF-β R2, and TGF-β R3)

Figure 2 displays RT-qPCR results for the gene expression of TGF-β1 and its receptors (TGF-β R1, TGF-β R2, and TGF-β R3). A significant decrease (*p* < 0.001) of about 70% was observed in the expression of TGF-β1, TGF-βR1, TGF-βR2, and TGF-βR3 when osteoblasts were treated with 50 μM zoledronate in the Col-M group if compared to untreated cells. However, these decreases in gene expression were not found when osteoblasts were treated with 50 μM zoledronate and after being cultured on the Geranyl-geraniol-doped membranes.

#### 3.2.2. Gene Expression of OPG-RANKL Complex

Figure 3 depicts the RT-qPCR results for the gene expression of RANKL, OPG and the OPG/RANKL ratio. Zolendronate-treated osteoblasts in the Col-M showed about an eight-fold (*p* < 0.001) decrease in the expression of RANKL, and a significant increase (*p* < 0.05) of about 30% in the expression of OPG compared to the untreated cells. However, this increase in OPG gene expression did not occur when osteoblasts were treated with 50 μM zoledronate in the Geranyl-geraniol (GG-M) group. Osteoblasts seeded onto the GG-M without zoledronate treatment also showed a significant decrease of about 75% in RANKL gene expression (*p* < 0.001). When considering the OPG/RANKL ratio, the addition of zoledronate up-regulates the ratio about 30 times in osteoblasts seeded on Col-M, if compared to the control group (*p* < 0.001). When zoledronate-treated osteoblasts were cultured on the GGOH-doped membrane, OPG/RANKL values were similar to those of the control group (*p* > 0.05). Non-treated cells cultured on GGOH-doped membranes attained lower OPG/RANKL ratio values than the control (about 1.5 lower) (*p* = 0.03).

#### 3.2.3. Effect of BPs on the Gene Expression of Runx2, ALP, Col-I, OSX, and OSC

The RT-qPCR results for the expression of osteoblast differentiation markers Runx2, ALP, Col-I, OSX, and OSC are displayed in Figure 4. In general, treatment of osteoblasts with 50 μM zoledronate in the Col-M attained a significant decrease in the expression of differentiation markers compared to the untreated cells. About a 2.5-fold decrease in the expression of Runx2, ALP, and OSX, a 1-fold decrease in the expression of COL-I, and a 3.5-fold decrease in the expression of OSC were encountered. When zoledronate-treated osteoblasts were cultured in the GG-M, decreases in gene expression were not observed, if compared to the control group. The expression of OSX was significantly up-regulated (about 50%; *p* < 0.001) when osteoblasts were seeded onto the GG-M, without zoledronate treatment.

#### 3.2.4. Gene Expression of BMP-2, BMP-7, and VEGF

The RT-qPCR results for the gene expression of BMP-2, BMP-7, and VEGF are reported in the Figure 5. The treatment of osteoblasts with zoledronate in the Col-M produced a significant decrease of about 60% in the expression of BMP-2 (*p* < 0.05) and about 75% in the expression of BMP-7 (*p* < 0.001), if compared to the untreated cells. When zoledronate-treated osteoblasts were seeded on the GG-M, the attained values for the gene expression of BMP-2, BMP-7, and VEGF were similar to those of the control group. The expression of VEGF was significantly up-regulated when osteoblasts were seeded onto GG-M membranes, attaining a 3-fold increase. When non-treated osteoblasts were cultured on GG-M, a 1-fold increase was produced.

### 3.3. Nodules Formation and Matrix Mineralization

The Alizarin Red staining results after 15 and 21 d of culture in osteogenic medium are shown in Figure 6. A significant decrease of about 25% in calcium deposition was observed when osteoblasts were treated with zoledronate and cultured on the undoped Col-M, if compared to untreated cells after 15 d of culture (*p* < 0.001), and a decrease of about 55% after 21 d (*p* < 0.001). Just slight decreases in calcium deposition were observed for zoledronate-treated osteoblasts in the geranylgeraniol-doped membranes after 15 d, and after 21 d, no differences were found if compared to the control group.

### 3.4. Scanning Electron Microscopy

Images from the FESEM analysis are displayed in Figure 7. When osteoblasts were cultured without zoledronate treatment, their morphology was spindle shaped and different layers of cells could be observed. In addition, mineral depositions and or extracellular matrices may also be found on the osteoblasts’ surfaces (Figure 7A–D).

In Figure 7E,F, osteoblasts seeded onto Col-M membranes with zoledronate treatment may be observed. The cells were flat with an undifferentiated morphology. A reduction in cell number is also patent.

When zoledronate-treated osteoblasts were seeded onto GG-M membranes (Figure 7G,H), the encountered cells were fusiform, and minerals deposits may be found on the osteoblasts’ and membranes’ surfaces.

## 4. Discussion

This study aimed to evaluate if collagen-based membranes doped with GGOH may help to reverse the negative impact of zoledronate on the growth and differentiation of human osteoblasts.

A pericardium-derived natural collagen membrane was selected for the present purpose as it provides a slow degradation rate based on: (i) the specific composition and structure of naturally cross-linked pericardial collagen fibers [35,36] and (ii) the natural multilayered and honeycomb-like collagen structure with a high content of collagen type III [36]. When used in the delivery of drug molecules, the slow rate of biodegradability of collagen membranes is important, as it may permit the production of a sustained and long-lasting effect [37].

The cell line selected for the assays was the MG-63, which is one of the most widely used cell lines to study osteoblast activity [8,34,38]. MG63 osteoblast-like cells have similar characteristics to primary human osteoblasts. In addition, MG63 cells have the advantage that they require a shorter isolation time and their availability is unlimited [38,39].

MTT assay was applied to study the proliferation of osteoblasts, followed by the study of the expression of proliferation-related genes using RT-qPCR (e.g., TGF-β1, TGFβ-R1, TGFβ-R2, or TGFβ-R3) [19]. Differentiation of osteoblasts was quantitatively tested by nodule formation and matrix mineralization analysis. Moreover, the expression of differentiation-related genes (e.g., ALP, OSC, BMP-2, BMP-7, Runx-2, or OSX) was also measured. The ratio between OPG and RANKL gene expression was analyzed with the aim of studying the bone remodeling capacity and the potential effect on bone mass preservation [40]. Finally, cell morphology was also analyzed by FESEM, as a previous association between changes in osteoblast morphology and its differentiation stages has been reported [41].

A significant decrease (about 60%) (*p* < 0.001) in cell proliferation was observed when osteoblasts were treated with zoledronate (50 µM) and cultured on the undoped membranes when compared to the untreated cells. Similar decreases in cell proliferation of about 55 and 35% have been previously described in osteoblasts exposed to 30 and 50 μM of zoledronate, respectively [42]. BPs produced cell cycle arrest and induced cell apoptosis/necrosis, consistent with BRONJ [17]. These effects have been attributed to the inhibition of farnesyl pyrophosphate (FPP) synthase, GGPP synthase, or both [42]. It has been previously shown that the induction of cell apoptosis is produced due to the formation of ATP analogues by suppressing the mevalonate pathway [43]. In the present study, zoledronate did not exert this detrimental effect on the osteoblasts grown on GG-M (*p* < 0.001) (Figure 1). It seems that GGOH can even rescue cells from the apoptotic effects of nitrogen-containing BPs. GGOH inhibits caspase-3 activation, and thus apoptosis, by inhibiting geranylgeranyl transferase I through an increase in the amount of substrate available for GTPase prenylation [43]. It has also been described that at low doses, GGOH (10–40 μM) may promote osteoblasts’ viability [25,43]. These results are in agreement with other authors who found that the treatment of osteoblasts with GGOH in combination with other BPs produced an attenuation of the detrimental effect of BPs on osteoblast viability [24,25,42].

The encountered changes on osteoblast proliferation are also consistent with the produced significant down-regulation of proliferative-related genes (TGF-β1, TGFβ-R1, TGFβ-R2, or TGFβ-R3) demonstrated by RT-qPCR in the zoledronate-treated osteoblasts which were cultured on undoped collagen membranes. This down-regulation was not produced when osteoblasts were treated with zoledronate on the GGOH-doped membranes (Figure 2). This can be visualized in the action on the TGF-β superfamily, which is composed of more than 40 members, including TGF-βs, Nodal, Activin, and BMPs. TGF-β acts as a key signal for the regulation of osteoblast proliferation, differentiation, and bone formation. It was previously described that moderately high levels of TGF-β1 result in stimulation of early osteoblast proliferation and that they are also involved in the activation of a mitogen-activated protein kinase (MAPK) cascade [44,45]. The beneficial effect of GGOH on the disruption of osteoblast activity could be attributed to an increase in the Rap1A/B protein which is suppressed by BPs. This fact results in changes in genes encoding for bone formation proteins, such as fibroblast growth factor-2, VEGF, COL-I, and osteopontin. In addition, VEGF exerts its action as a modulator of bone repairing, promoting osteoblast maturation/differentiation and angiogenesis [43].

VEGF merits special attention as it is a signaling pro-angiogenic protein affecting endothelial cell growth, migration, and vessel formation in many tissues. It regulates vascular growth in the skeleton [46,47]. It is produced by osteoblasts and osteocytes, among other bone cells [48,49]. Zolendronate produced a significant down-regulation on VEGF expression of about three times (Figure 5). The anti-angiogenic effect of BPs has been shown to play a crucial role in the pathogenesis of BRONJ, the combination of infection and inflammation caused by bone trauma, suppressed bone remodeling by anti-resorptive agents, and inhibition of new blood vessel formation by anti-angiogenic agents [43]. BPs inhibit VEGF expression by acting on the mevalonate pathway [50]; therefore, it was expected that GGOH may help to reverse this effect. When zoledronate-treated osteoblasts were seeded onto GGOH membranes, the expression of VEGF was significantly up-regulated more than 15 times. It is interesting that GGOH may also act as a pro-angiogenic compound, regardless of zoledronate treatment, as it was able to increase VEFG production in non-treated osteoblasts about three times.

COL-I was down-regulated about six times after zoledronate treatment of osteoblasts. Collagen depletion and damage has also been proposed as an important mechanism contributing to the poor tissue integrity observed in BRONJ [42]. In the present research, GGOH half restored the produced decrease in COL-I gene expression.

The osteogenic differentiation of osteoblasts is marked by the expression of some proteins and transcription factors such as OSX, OPG, and ALP [51,52]. COL-I and OSC act as osteogenic markers of late osteoblast differentiation that appear at the onset of mineralization. Among them, OSX is a key transcription factor for bone formation and the osteoblast differentiation process. When expressed, bipotentiality from preosteoblast to osteoblast and chondrocyte is lost. In addition, BMPs have different effects on osteoblast differentiation, proliferation and, in general, on osteoblast physiology. Among them, BMP-2 and BMP-7 exert an important role in osteoblast differentiation and bone remodeling/formation by inducing the expression of some osteoblastic markers like ALP, thus promoting mineralization [53,54]. All the tested osteogenic genes related to bone development (ALP, OSX, RUNX2, OSC, BMP2, BMP7, and VEGF) were significantly down-regulated after zoledronate treatment of osteoblasts cells (50 µM) (Figure 4 and Figure 5). The highly significant down-regulation for some of these genes by zoledronate may point out the alteration of important tissue repair regulatory pathways. During injury or wound repair, bone growth factors play a crucial role, and the down-regulation of these genes may, therefore, contribute to the pathogenesis of BRONJ [43]. In the present research, GGOH doped onto collagen membranes was able to always restore gene expression. [42], demonstrating the alteration of a high number of genes after zoledronate treatment, but the co-addition of GGOH (50 µM) with ZA (30 µM) in the gene expression assays only showed statistically significant correction of four of the previously down-regulated genes. The major success of the present treatment may be due to the lower applied dose of GGOH (10 µM) and to the adopted releasing strategy, as it was doped onto slow-degrading collagen membranes.

Special attention should be paid to the OPG/RANKL ratio. When osteoblasts were cultured on undoped membranes, zoledronate up-regulated the ratio up to 30 times, if compared to the control group (Figure 3). The effect of GGOH on osteoclasts in vitro remains unclear. Some studies have suggested that GGOH interferes with the signaling pathways of osteoclast formation and activity which might have an inhibitory effect by affecting certain proteins and enzymes involved in their regulation [55,56]. Others authors have demonstrated opposite results by showing that GGOH promotes osteoclast activity and proliferation at doses between 50 and 100 µM [43,57]. In the presence of BPs, GGOH antagonises the suppressive effect of BPs on osteoclasts. GGOH also prevents the loss of actin rings and nucleus condensation produced by alendronate and risedronate [58,59]. It is one of the main actions of BPs in bone, acting as an anti-resorptive drug [43]. However, it should be noted that GGOH, in vitro, completely reversed the activity of BPs on the OPG/RANKL ratio. This is of importance as it may indicate that GGOH systemic administration may antagonize the beneficial effects of BPs in treated patients. Knowing that GGOH is a natural compound present in in several oils and vegetables [60], and even when the bioavailability of the various GGOH derivatives or the amount of GGOH that must be absorbed to suppress the activity of BPs are not known, it has now been recognized as a safe and widely available dietary supplement [43]. After the present results, it is recommended that its use has to be validated in clinical trials to show its efficacy in mitigating BRONJ in humans, without counteracting the systemic effect of BPs.

Differentiation of osteoblasts was also quantitatively tested by the study of nodule formation and matrix mineralization. A significant decrease in calcium deposition was observed when osteoblasts were treated with zoledronate in the Col-M group compared to the untreated cells at 15 or 21 days of culture (*p* < 0.001) (Figure 6). However, the reported difference in calcium deposition was diminished when osteoblasts were treated with zoledronate in the GG-M group at 15 d of culture, and completely reversed after 21 d. These results comply with previous results where BPs inhibited the differentiation of osteoblasts and decreased mineralization [18,25,42], an effect that was attenuated by the addition of GGOH to the culture medium [25,42]. The action mechanism has also been attributed to reestablishment of the mevalonate pathway [25].

According to the proliferation and gene expression results, after FESEM analysis, zoledronate-treated osteoblast cells were scarce, flat, and with an undifferentiated appearance when cultured onto Col-M membranes. Those osteoblasts seeded onto GGOH-doped collagen membranes were fusiform and exhibited abundant extracellular matrices or mineral deposits (Figure 7), regardless of zoledronate treatment.

Previous research has been carried out studying the effect of GGOH on osteoblasts, differing from our research, in that it was directly applied in the cell culture medium [24,42]. Some different effects have been encountered, but a positive effect of GGOH application has been reported for BPs-treated osteoblasts. Exceptions with high-dose BPs can be found because non-mevalonate pathways can be activated; this will induce cells apoptosis that cannot be restored by GGOH [43]. In vivo exposure to GGOH at concentrations below 50 μM appears to be safe, and toxicity data for GGOH are scarce. However, delivery of GGOH at a precise dose through a slow-releasing strategy like nanomaterials is recommended [43].

In the present research, GGOH is doped in collagen membranes for GBR, providing a modification of the collagen membranes for preventing BRONJ which is biomimetical, non-toxic, and economic, in addition to being easy to obtain. Collagen biomimetic membranes, in particular, types I and III, have multiple advantages such as haemostatic property, weak immunogenicity, chemotactic action for fibroblasts, and osteoblastic adhesion; therefore, they are of great interest in GBR/GTR [61]. Drug delivery systems are defined as a device or formulation capable of releasing an active substance to a target tissue, increasing the efficacy of the active substance [62]. Achieving drug release from a resorbable collagen membrane may be considered a bio-inspired design, as in biological systems, tissue healing occurs via chemical releases at the site of damage, which afterwards, initiate a systemic response to transport repairing agents to the healing site. Polymeric membranes, such as those used in this study, can be used as drug carriers. In this way, pharmacological activity is increased, and drug side effects are reduced.

The results obtained in our study can be considered as a starting point for pre-clinical and clinical research to further investigate the ability of GGOH to reverse the effect of BPs in patients treated with this antiresorptive medication who need to undergo GBR/GTR procedures with collagen membranes.

A limitation of the present research is that it is an in vitro study; therefore, preclinical in vivo investigations are needed in order to confirm the obtained results. Two important points should be considered to encourage developing GGOH-doped membranes as an interventional agent for BRONJ: (i) mainly due to the lack of evidence related to BRONJ pathogenesis, there is no definitive treatment to date; (ii) topical application of GGOH, in contrast to systemic, should be preferred as it will require lower dose ranges, and systemic application might antagonize the therapeutic effects of BPs in treated patients [43].

Therefore, in order to develop an effective treatment strategy for BRONJ, a clearer understanding of BPs-induced cell behavior in the oral cavity is necessary [42]. Future investigation of the method of delivery, precise dosing, and mechanisms of action should also be undertaken both pre-clinically and clinically. Future strategies should include the application of these membranes in animal models in order to test whether their use as a carrier for GGOH is useful for the prevention of BRONJ, when GBR procedures are necessary.

## 5. Conclusions

In conclusion, the use of collagen membranes doped with GGOH is able to reverse the negative impact of zoledronate on the proliferation, differentiation, and gene expression of several osteoblast markers. For this reason, and with the lack of further research, GGOH doping of collagen membranes could be considered as a potential therapeutic resource for the treatment of patients consuming BPs and needing to undergo GBR/GTR interventions.

## Figures and Tables

**Figure 1 biomimetics-09-00004-f001:**
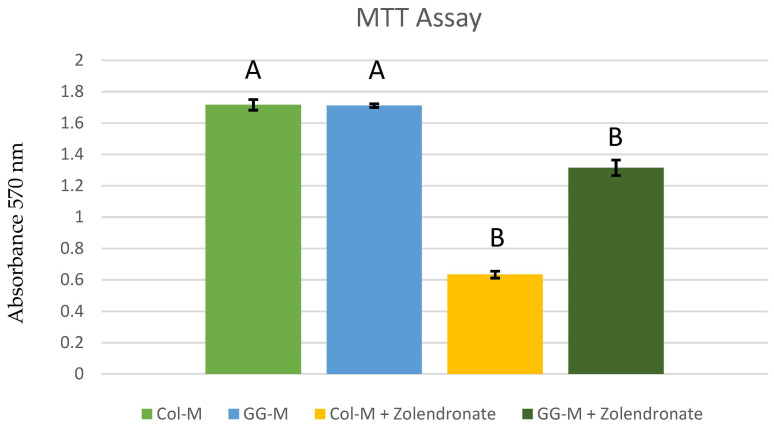
Osteoblast proliferation after 48 h of culture onto the different doped membranes. Data are expressed as mean and standard deviation. Significant differences are indicated by different letters after multiple comparisons (*p* < 0.05).

**Figure 2 biomimetics-09-00004-f002:**
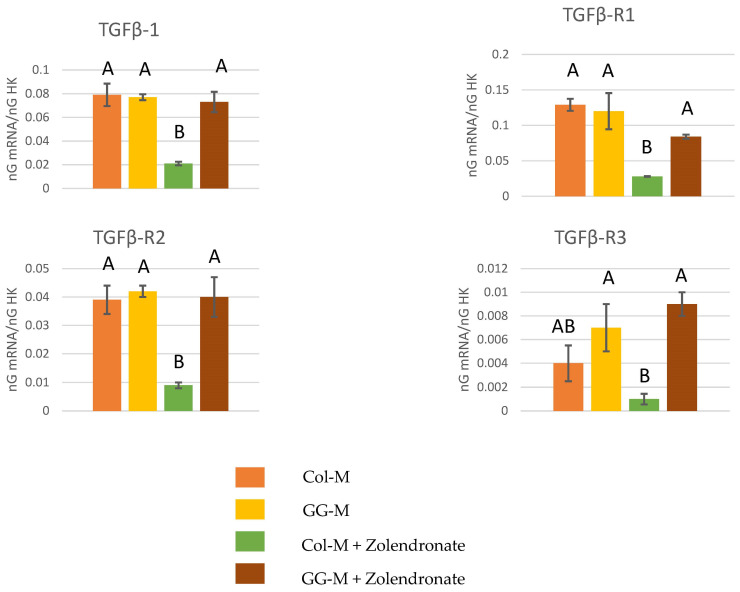
Gene expression analysis of TGF-β1, TGF-βR1, TGF-βR2 and TGF-βR3 by real-time PCR for osteoblasts seeded onto the different experimental membranes after 48 h of culture. Mean and standard deviation of ng mRNA/ngHK is presented for each experimental group. Significant differences are indicated by different letters, after multiple comparisons (*p* < 0.05).

**Figure 3 biomimetics-09-00004-f003:**
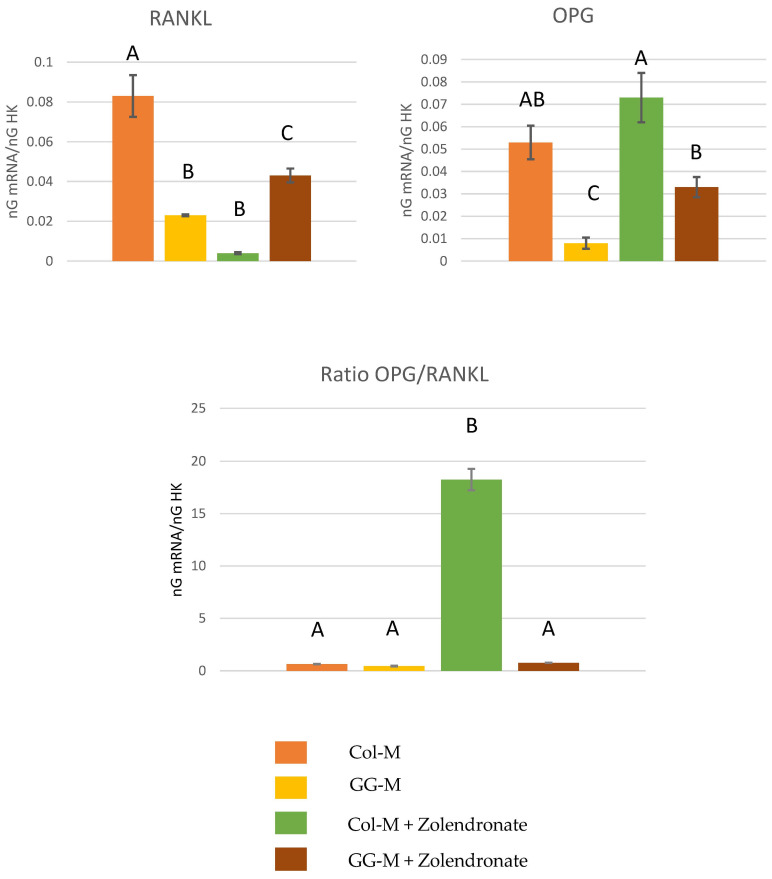
Gene expression analysis of RANKL, OPG and OPG/RANKL by real-time PCR for osteoblasts seeded onto the different experimental membranes after 48 h of culture. Mean and standard deviation of ng mRNA/ngHK is presented for each experimental group. Significant differences are indicated by different letters, after multiple comparisons (*p* < 0.05).

**Figure 4 biomimetics-09-00004-f004:**
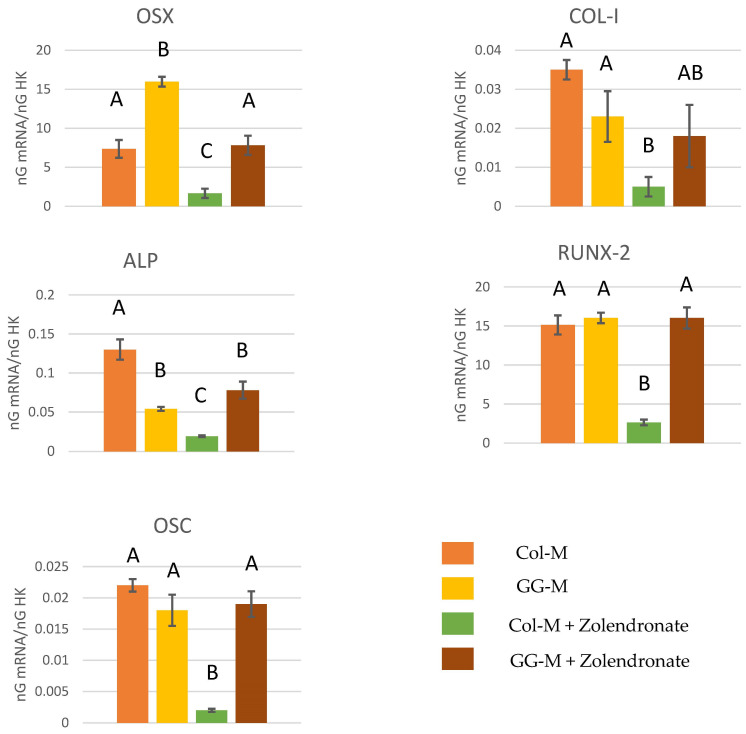
Gene expression analysis of OSX, COL-I, ALP, RUNX-2, and OSC by real-time PCR for osteoblasts seeded onto the different experimental membranes after 48 h of culture. Mean and standard deviation of ng mRNA/ngHK is presented for each experimental group. Significant differences are indicated by different letters, after multiple comparisons (*p* < 0.05).

**Figure 5 biomimetics-09-00004-f005:**
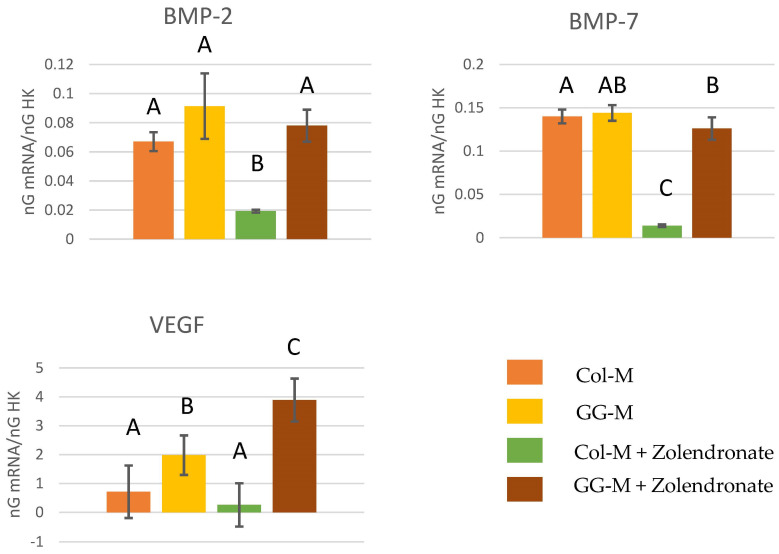
Gene expression analysis of BMP-2, BMP-7, and VEGF by real-time PCR for osteoblasts seeded onto the different experimental membranes after 48 h of culture. Mean and standard deviation of ng mRNA/ngHK is presented. Significant differences are indicated by different letters after multiple comparisons (*p* < 0.05).

**Figure 6 biomimetics-09-00004-f006:**
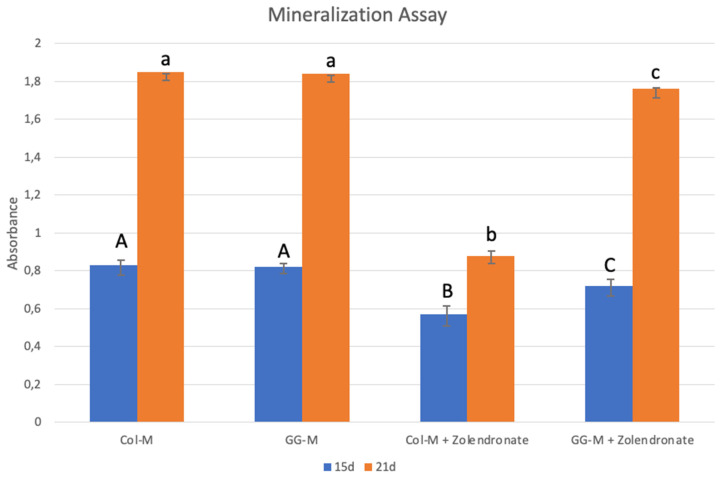
Mean and standard deviation of absorbance obtained after culture of the osteoblasts onto the different doped membranes for 15 and 21 days using the Alizarin Red S method. Data are expressed as mean and standard deviation. Different letters indicate significant difference after one-way ANOVA and post hoc multiple comparisons (*p* < 0.05).

**Figure 7 biomimetics-09-00004-f007:**
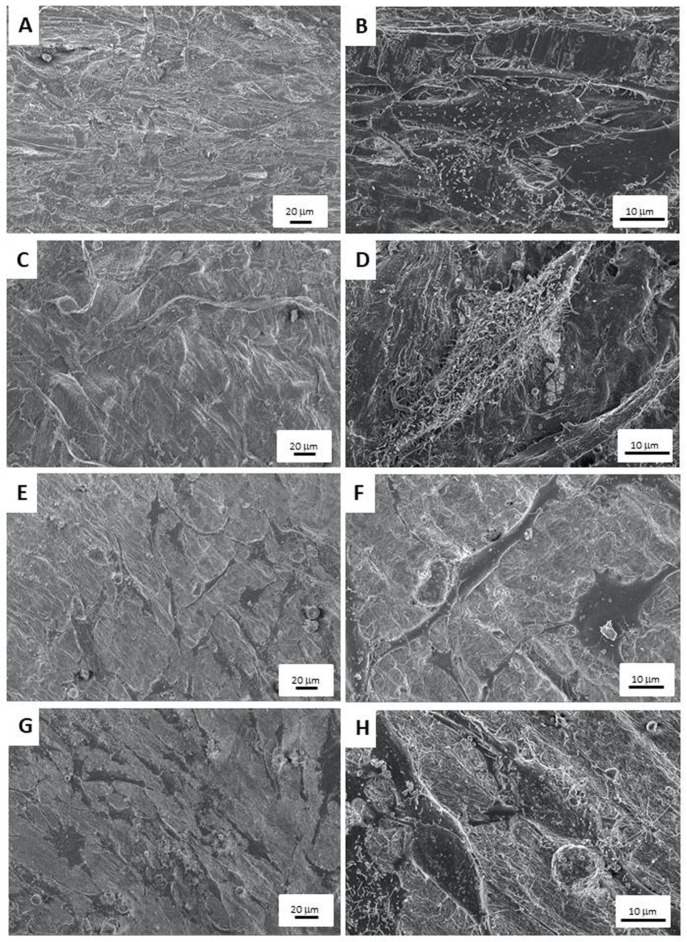
FESEM images of: (**A**,**B**) osteoblasts cultured onto collagen membranes, (**C**,**D**) osteoblasts seeded onto GGOH-doped membranes, (**E**,**F**) zoledronate-treated osteoblasts cultured onto collagen membranes, and (**G**,**H**) zoledronate-treated osteoblasts seeded onto GGOH-doped membranes. Images were taken at 3 kV and distances were between 3.7 and 4.1 mm. Magnifications at images are 300× and 1500×.

**Table 1 biomimetics-09-00004-t001:** Sense and antisense primer sequences for real-time qPCR.

Gene	Sense Primer (5′–3′)	Antisense Primer
ALP	CCAACGTGGCTAAGAATGTCATC	TGGGCATTGGTGTTGTACGTC
BMP-2	TCGAAATTCCCCGTGACCAG	CCACTTCCACCACGAATCCA
BMP-7	CTGGTCTTTGTCTGCAGTGG	GTACCCCTCAACAAGGCTTC
Col-1	AGAACTGGTACATCAGCAAG	GAGTTTACAGGAAGCAGACA
OSX	TGCCTAGAAGCCCTGAGAAA	TTTAACTTGGGGCCTTGAGA
OPG	ATGCAACACAGCACAACATA	GTTGCCGTTTTATCCTCTCT
OSC	CCATGAGAGCCCTCACACTCC	GGTCAGCCAACTCGTCACAGTC
RANKL	ATACCCTGATGAAAGGAGGA	GGGGCTCAATCTATATCTCG
Runx-2	TGGTTAATCTCCGCAGGTCAC	ACTGTGCTGAAGAGGCTGTTTG
TGFβ1	TGAACCGGCCTTTCCTGCTTCTCATG	GCGGAAGTCAATGTACAGCTGCCGC
TGFβ-R1	ACTGGCAGCTGTCATTGCTGGACCAG	CTGAGCCAGAACCTGACGTTGTCATATCA
TGFβ-R2	GGCTCAACCACCAGGGCATCCAGAT	CTCCCCGAGAGCCTGTCCAGATGCT
TGFβ-R3	ACCGTGATGGGCATTGCGTTTGCA	GTGCTCTGCGTGCTGCCGA TGCTGT
VEGF	CCTTGCTGCTCTACCTCCAC	CACACAGGATGGCTTGAAGA
UBC	TGGGATGCAAATCTTCGTGAAGACCCTGAC	ACCAAGTGCAGAGTGGACTCTTTCTGGATG
PPIA	CCATGGCAAATGCTGGACCCAACACAAATG	TCCTGAGCTACAGAAGGAATGATCTGGTGG
RPS13	GGTGTTGCACAAGTACGTTTTGTGACAGGC	TCATATTTCCAATTGGGAGGGAGGACTCGC

## Data Availability

The data presented in this study are available on request from the corresponding author.

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
