# Peer review of "Biomimetic Collagen Membranes as Drug Carriers of Geranylgeraniol to Counteract the Effect of Zoledronate"

_biomimetics, 2023, doi:10.3390/biomimetics9010004_

Round 1
Reviewer 1 Report
Comments and Suggestions for Authors
This manuscript demonstrates the capacity of collagen-based membranes doped with geranylgeraniol to reverse any negative effect of zoledronate on the growth and differentiation of human osteogenic cells. The use of biocompatible membranes that mimic the microenvironment of ECM has gained attention by many researchers recently. Hence, the proposed topic of this study seems to be meaningful. However, there are some issues related to this manuscript, such as poor readability and confusing experimental setup. Therefore, it is recommended to address the concerns listed below to improve the quality of this manuscript.
1. Overall, readability (writing) can be improved. It is difficult to follow what the authors aim to express. Also, please emphasize why this study is important and innovative.
2. In some sections, indents and line spacings are not uniform.
3. Page 4, does FBS stand for Fetal Bovine Serum or Foetal Bovine Serum?
4. Is there any specific reason that the authors only choose not multiple, but a single time point (48 h) for MTT assay, PCR, and SEM? If so, please explain.
5. It is acceptable to use Excel to draw graphs. However, it is recommended to use a professional tool (Origin or others) if possible.
6. In Figure 2, the alignment of the graphs needs to be improved.
7. In Figure 3, can the authors create a separate graph that magnifies the control, geranil, and geranil+ zoledronate groups in Ratio OPG/RANKL?
8. In Figure 6, it would look better to compare the data if both graphs could be merged.
9. Be precise and brief in discussion. There is no need to repeat the sentences that are mentioned in the introduction or result sections.
10. Page 15, (figure 1) -> (Figure 1), capitalization.
Author Response
Dear Editor,
We are pleased to send you a new version of our manuscript: biomimetics-2727393 "Biomimetic collagen membranes as drug-carriers of geranylgeraniol to counteract the effect of zoledronate" by Francisco Javier Manzano-Moreno, Elvira De Luna-Bertos*, Manuel Toledano-Osorio, Paula Urbano-Arroyo, Concepción Ruiz, Manuel Toledano and Raquel Osorio.
We are very grateful for the comments and insights of your reviewers, which allowed us to strengthen and clarify our paper. We trust that this new version of the article is now suitable for publication in Biomimetics.
We look forward to your decision.
Sincerely yours,
Address for correspondence:
Elvira de Luna-Bertos
Department of Nursing, Faculty of Health Sciences, University of Granada, Spain.
Avenida de la Ilustración s/n, 18016-Granada, Spain.
e-mail: elviradlb@ugr.es
RESPONSES TO #REVIEWER 1:
- Concern of the reviewer: Overall, readability (writing) can be improved. It is difficult to follow what the authors aim to express. Also, please emphasize why this study is important and innovative.
Our response: Thank you for your comment. We have emphasized why this study is important and innovative in the introduction and discussion sections.
Revised text:
Lines 103-106: So far there are no studies analysing the effect of doping collagen membranes with GGOH on bisphosphonate-treated osteoblasts. This fact can be considered very relevant because there are many patients treated with bisphosphonates who will need to undergo GBR/GTR procedures for oral rehabilitation.
Lines 574-577: The results obtained in our study can be considered as a starting point for pre-clinical and clinical research to further investigate the ability of GGOH to reverse the effect of BPs in patients treated with this antiresorptive medication who need to undergo GBR/GTR procedures with collagen membranes.
- Concern of the reviewer: In some sections, indents and line spacings are not uniform.
Our response: We have corrected indents and line spacing.
Revised text: Not applicable
- Concern of the reviewer: Page 4, does FBS stand for Fetal Bovine Serum or Foetal Bovine Serum?
Our response: Thank you for your correction. The correct word is Fetal. It has been changed in the text.
Revised text: Line 128: “…fetal…”
- Concern of the reviewer: Is there any specific reason that the authors only choose not multiple, but a single time point (48 h) for MTT assay, PCR, and SEM? If so, please explain.
Our response: In most of the articles in the literature, this assay is performed between 24 and 48 hours after cell treatment, because after longer periods of time the osteoblast loses its proliferative capacity, and begins to differentiate once it adheres to the culture surface. Gene expression analysis (PCR) can be performed at different times. However, in order to follow a continuity with the studies of cell viability, we decided to perform the PCR study at 48 hours of culture, enough time to show changes in gene expression of markers of cell differentiation and proliferation, as shown in previous studies in the literature.
Revised text: Not applicable
Some references about this point are in the manuscript:
- Manzano-Moreno, F.J.; Ramos-Torrecillas, J.; De Luna-Bertos, E.; Ruiz, C.; García-Martínez, O. High Doses of Bisphosphonates Reduce Osteoblast-like Cell Proliferation by Arresting the Cell Cycle and Inducing Apoptosis. J Craniomaxillofac Surg 2015, 43, 396–401, doi:10.1016/j.jcms.2014.12.008.
- Manzano-Moreno, F.J.; Ramos-Torrecillas, J.; De Luna-Bertos, E.; Reyes-Botella, C.; Ruiz, C.; García-Martínez, O. Nitrogen-Containing Bisphosphonates Modulate the Antigenic Profile and Inhibit the Maturation and Biomineralization Potential of Osteoblast-like Cells. Clin Oral Investig 2015, 19, 895–902, doi:10.1007/s00784-014-1309-z.
- Manzano-Moreno, F.J.; Ramos-Torrecillas, J.; Melguizo-Rodríguez, L.; Illescas-Montes, R.; Ruiz, C.; García-Martínez, O. Bisphosphonate Modulation of the Gene Expression of Different Markers Involved in Osteoblast Physiology: Possible Implications in Bisphosphonate-Related Osteonecrosis of the Jaw. Int J Med Sci 2018, 15, 359–367, doi:10.7150/ijms.22627.
- Otto, M.; Lux, C.; Schlittenbauer, T.; Halling, F.; Ziebart, T. Geranyl-Geraniol Addition Affects Potency of Bisphosphonates-a Comparison in Vitro Promising a Therapeutic Approach for Bisphosphonate-Associated Osteonecrosis of the Jaw and Oral Wound Healing. Oral Maxillofac Surg 2022, 26, 321–332, doi:10.1007/s10006-021-00982-8.
- Zafar, S.; Coates, D.E.; Cullinan, M.P.; Drummond, B.K.; Milne, T.; Seymour, G.J. Effects of Zoledronic Acid and Geranylgeraniol on the Cellular Behaviour and Gene Expression of Primary Human Alveolar Osteoblasts. Clin Oral Investig 2016, 20, 2023–2035, doi:10.1007/s00784-015-1706-y.
- Concern of the reviewer: It is acceptable to use Excel to draw graphs. However, it is recommended to use a professional tool (Origin or others) if possible.
Our response: Authors very much appreciate the reviewer's comment. Figures quality has been augmented.
Revised text: Not applicable
- Concern of the reviewer: In Figure 2, the alignment of the graphs needs to be improved.
Our response: Thank you for your suggestion. We have corrected the alignment of the figure 2.
Revised text: Please, see the revised figure 2.
- Concern of the reviewer: In Figure 3, can the authors create a separate graph that magnifies the control, geranil, and geranil+ zoledronate groups in Ratio OPG/RANKL?
Our response: Thank you for your comment. We have modified the graph to magnify the different groups.
Revised text: Please, see the revised figure 3.
- Concern of the reviewer: In Figure 6, it would look better to compare the data if both graphs could be merged.
Our response: Thank you for your suggestion. We have merged both graphs.
Revised text: Please, see the new figure 6.
- Concern of the reviewer: Be precise and brief in discussion. There is no need to repeat the sentences that are mentioned in the introduction or result sections.
Our response: We have completely revised the discussion and tried to accomplish the reviewer suggestion.
Revised text: Please, see the revised version of the discussion section.
- Concern of the reviewer: Page 15, (figure 1) -> (Figure 1), capitalization.
Our response: It has been modified in the text.
Revised text: Line 443: (Figure 1).
Reviewer 2 Report
Comments and Suggestions for Authors
In this study, Manzano-Moreno et al. investigated the capacity of biomimetic collagen membranes as drug carriers of the diterpenoid geranylgeraniol (GGOH). The authors utilized collagen membranes as the loading material for the poorly water-soluble GGOH, and examined influence of GGOH on the negative effect of bisphosphpnates on osteoblasts. The background of this study was well-described, the experimental system is simple and straightforward, and discussion was carefully elaborated. However, this manuscript has several concerns that require the authors’ attention.
1. The authors chose a dose of zoledronate 50 uM in this study, although the rationale is unclear. The authors need to add experimental results to explain why 50 uM was chosen or describe the rationale for selecting this concentration clearly.
2. Throughout this study, the description of figure legends was not clear and confusing: Control, Geranil, Zolendronate, and Geranil+Zolendronate. For example, on page 8, the authors stated, “Zoledronate treated osteoblasts in the Col-M showed a four-fold decrease expression of RANKL”. However, in the corresponding figure (Fig3A), Zoledronate-treated group showed more than a one tenth reduction. Since the authors referred to undoped collagen membranes as Col-M and GGOH-doped collagen membranes as GC-M in the Methods and Results sections, it would be better to use these consistent terms in the figure legends as well. The description of the results should also be revised accordingly.
3. As the authors discussed in the discussion, the results of OPG/RANKL ratio after treatment with GGOH was quite impressive, and strongly imply the effect of GGOH on osteoclasts, too. It would be better to add additional discussion about the effect of GGOH on osteoclasts and bone homeostasis.
Author Response
Dear Editor,
We are pleased to send you a new version of our manuscript: biomimetics-2727393 "Biomimetic collagen membranes as drug-carriers of geranylgeraniol to counteract the effect of zoledronate" by Francisco Javier Manzano-Moreno, Elvira De Luna-Bertos*, Manuel Toledano-Osorio, Paula Urbano-Arroyo, Concepción Ruiz, Manuel Toledano and Raquel Osorio.
We are very grateful for the comments and insights of your reviewers, which allowed us to strengthen and clarify our paper. We trust that this new version of the article is now suitable for publication in Biomimetics.
We look forward to your decision.
Sincerely yours,
Address for correspondence:
Elvira de Luna-Bertos
Department of Nursing, Faculty of Health Sciences, University of Granada, Spain.
Avenida de la Ilustración s/n, 18016-Granada, Spain.
e-mail: elviradlb@ugr.es
RESPONSES TO #REVIEWER 2:
In this study, Manzano-Moreno et al. investigated the capacity of biomimetic collagen membranes as drug carriers of the diterpenoid geranylgeraniol (GGOH). The authors utilized collagen membranes as the loading material for the poorly water-soluble GGOH, and examined influence of GGOH on the negative effect of bisphosphpnates on osteoblasts. The background of this study was well-described, the experimental system is simple and straightforward, and discussion was carefully elaborated. However, this manuscript has several concerns that require the authors’ attention.
- Concern of the reviewer: The authors chose a dose of zoledronate 50 µM in this study, although the rationale is unclear. The authors need to add experimental results to explain why 50 µM was chosen or describe the rationale for selecting this concentration clearly.
Our response: Thank you for your question. On the basis of previous studies by our group and the existing literature, the dose of 50 µM is reported as a therapeutic dose. Doses above 50 µM have shown a toxic effect on the osteoblast [18,19,24]. We have included a new sentence in Material and Methods to clarify this point.
Revised text: Line 205: “…(which is reported as a therapeutic dose [18,19,24])…”
- Concern of the reviewer: Throughout this study, the description of figure legends was not clear and confusing: Control, Geranil, Zolendronate, and Geranil+Zolendronate. For example, on page 8, the authors stated, “Zoledronate treated osteoblasts in the Col-M showed a four-fold decrease expression of RANKL”. However, in the corresponding figure (Fig3A), Zoledronate-treated group showed more than a one tenth reduction. Since the authors referred to undoped collagen membranes as Col-M and GGOH-doped collagen membranes as GC-M in the Methods and Results sections, it would be better to use these consistent terms in the figure legends as well. The description of the results should also be revised accordingly.
Our response: Thank you for your suggestion. We apologize for the mistake. Zolendronate treated osteoblasts in the Col-M showed about an eight-fold decrease expression of RANKL. We have corrected it in the text. We have modified the figure legends to be consistent with the text.
Revised text: Lines 256-257: “Zolendronate treated osteoblasts in the Col-M showed about an eight-fold (p<0.001) decrease expression of RANKL”
Please, see the revised figure legends.
- Concern of the reviewer: As the authors discussed in the discussion, the results of OPG/RANKL ratio after treatment with GGOH was quite impressive, and strongly imply the effect of GGOH on osteoclasts, too. It would be better to add additional discussion about the effect of GGOH on osteoclasts and bone homeostasis.
Our response: Thank you for your suggestion. We have added new information about the effect of GGOH on osteoclasts in the discussion section.
Revised text: Lines 515-523: “The effect of GGOH on osteoclasts in vitro remains unclear. Some studies have suggested that GGOH interferes with the signaling pathways of osteoclast formation and activity that might have an inhibitory effect by affecting certain proteins and enzymes involved in their regulation [55,56]. Others authors have demonstrated opposite results by showing that GGOH promotes osteoclasts activity and proliferation at doses between 50-100 µM [43,57]. In the presence of BPs, GGOH antagonises the suppressive effect of BPs on the osteoclast. GGOH also prevents the loss of actin rings and nucleus condensation produced by alendronate and risedronate [58,59].”
References:
- Ho, H.-J.; Shirakawa, H.; Giriwono, P.E.; Ito, A.; Komai, M. A Novel Function of Geranylgeraniol in Regulating Testosterone Production. Biosci Biotechnol Biochem 2018, 82, 956–962, doi:10.1080/09168451.2017.1415129.
- Hiruma, Y.; Nakahama, K.; Fujita, H.; Morita, I. Vitamin K2 and Geranylgeraniol, Its Side Chain Component, Inhibited Osteoclast Formation in a Different Manner. Biochem Biophys Res Commun 2004, 314, 24–30, doi:10.1016/j.bbrc.2003.12.051.
- van Beek, E.; Pieterman, E.; Cohen, L.; Löwik, C.; Papapoulos, S. Farnesyl Pyrophosphate Synthase Is the Molecular Target of Nitrogen-Containing Bisphosphonates. Biochem. Biophys. Res. Commun. 1999, 264, 108–111, doi:10.1006/bbrc.1999.1499.
- Fisher, J.E.; Rogers, M.J.; Halasy, J.M.; Luckman, S.P.; Hughes, D.E.; Masarachia, P.J.; Wesolowski, G.; Russell, R.G.; Rodan, G.A.; Reszka, A.A. Alendronate Mechanism of Action: Geranylgeraniol, an Intermediate in the Mevalonate Pathway, Prevents Inhibition of Osteoclast Formation, Bone Resorption, and Kinase Activation in Vitro. Proc Natl Acad Sci U S A 1999, 96, 133–138, doi:10.1073/pnas.96.1.133.
- Reszka, A.A.; Halasy-Nagy, J.M.; Masarachia, P.J.; Rodan, G.A. Bisphosphonates Act Directly on the Osteoclast to Induce Caspase Cleavage of Mst1 Kinase during Apoptosis. A Link between Inhibition of the Mevalonate Pathway and Regulation of an Apoptosis-Promoting Kinase. J Biol Chem 1999, 274, 34967–34973, doi:10.1074/jbc.274.49.34967.
Round 2
Reviewer 2 Report
Comments and Suggestions for Authors
The authors responded all my concerns appropriately. I'm satisfied.